# Long-Term Clinical Outcomes and Safety Analysis of Superficial Esophageal Cancer Patients Treated with Definitive or Adjuvant Radiotherapy

**DOI:** 10.3390/cancers14143423

**Published:** 2022-07-14

**Authors:** Bo Lyu, Yutian Yin, Yilin Zhao, Xu Yang, Jie Gong, Mai Zhang, Guangjin Chai, Zhaohui Li, Mei Shi, Zhouguang Hui, Lina Zhao

**Affiliations:** 1Department of Radiation Oncology, Xijing Hospital, Air Force Medical University, Xi’an 710000, China; doctor_lvbo@163.com (B.L.); yinyutian1123@126.com (Y.Y.); 18700182707@163.com (J.G.); radzhangmai@foxmail.com (M.Z.); doctoraxb@163.com (G.C.); qdds2007@163.com (Z.L.); mshi82@foxmail.com (M.S.); 2Department of Clinical Oncology, Xijing Hospital, Air Force Medical University, Xi’an 710000, China; yvette1027@163.com; 3Department of Radiation Oncology, National Cancer Center/National Clinical Research Center for Cancer/Cancer Hospital, Chinese Academy of Medical Sciences and Peking Union Medical College, Beijing 100000, China; yx1016534625@163.com; 4Department of VIP Medical Services, National Cancer Center/National Clinical Research Center for Cancer/Cancer Hospital, Chinese Academy of Medical Sciences and Peking Union Medical College, Beijing 100000, China

**Keywords:** superficial esophageal cancer, radiotherapy, outcomes research, toxicities

## Abstract

**Simple Summary:**

For superficial esophageal cancer, many tumor features have been identified as risk factors for lymph node metastasis, including depth of invasion, tumor size, the presence of lymphovascular invasion, etc. In such cases, endoscopic resection (ER) alone is not adequate for treatment coverage, and additional radiotherapy (RT)/concurrent chemoradiotherapy (CCRT) merits further investigation. On the other hand, esophagectomy has a high cure rate for early stage esophageal squamous cell cancer (ESCC); however, there are concerns of treatment complications and loss of esophagus preservation. Hence, definitive RT/CCRT may be a promising treatment alternative to esophagectomy, especially for those who are unable or unwilling to receive surgery. We retrospectively analyzed outcomes of patients with T1N0M0 staged ESCC treated with ER + RT/CCRT or RT/CCRT. We conclude that definitive or adjuvant RT/CCRT is an effective treatment alternative for superficial ESCC patients with satisfactory clinical outcomes and acceptable toxicities.

**Abstract:**

(1) Background: The role of radiotherapy (RT) in superficial esophageal squamous cell cancer (ESCC) remains unclear. The objective of our study was to perform a detailed outcome and safety analysis of RT as a definitive or adjuvant treatment for T1N0M0 staged ESCC patients. (2) Methods: A total of 55 patients treated with endoscopic resection (ER) + RT/concurrent chemoradiotherapy (CCRT) or RT/CCRT from January 2011 to June 2021 were included in this study. Eighteen patients with risk factors received ER + RT/CCRT, and thirty-seven patients solely received RT/CCRT. Kaplan–Meier curves were used to calculate the clinical outcomes, and toxicities were scored. (3) Results: The median follow-up time was 51.9 months. The estimated 5-year local recurrence-free survival (LRFS) and overall survival (OS) were 88.9% and 94.4% in the ER + RT/CCRT group and 91.8% and 91.7% in the RT/CCRT group. The predominant failure pattern was in-field local failure (5.5%, 3/55), with one patient in the ER + RT/CCRT group and two patients in the RT/CCRT group. One patient (1.8%, 1/55) had lung metastasis in the RT/CCRT group. The most common toxicities were Grades 1–2 in all patients, including esophagitis (74.5%, 41/55), myelosuppression (49.1%, 27/55) and esophageal stricture after RT (27.3%, 15/55). Two patients (11.1%, 2/18) and four patients (10.8%, 4/37) had Grade 3 esophageal stricture after RT in the ER + RT/CCRT group and RT/CCRT group, respectively. No patients experienced a Grade 4 or higher toxicity, and there were no treatment-related deaths. (4) Conclusions: Definitive or adjuvant RT/CCRT is an effective treatment alternative for superficial ESCC patients with satisfactory clinical outcomes and acceptable toxicities.

## 1. Introduction

Superficial esophageal cancer, also referred to as early stage esophageal cancer, is defined as having tumor invasion no deeper than the submucosa. The incidence of superficial esophageal cancer has increased in recent decades, especially in Asia, where endoscopic surveillance or screening has become more common [1,2,3]. Mucosal tumors are divided into M1, M2 and M3 and submucosa tumors are classified into SM1, SM2 and SM3 based on the depth of invasion. M1 tumors correspond to the Tis stage, M2 and M3 tumors are akin to T1a and SM1, SM2 and SM3 can be considered T1b, according to the American Joint Committee on Cancer (AJCC).

Endoscopic approaches are increasingly used in superficial esophageal cancer patients, especially in the T1a stage [4,5]. The most common surgical method is endoscopic resection (ER), including endoscopic submucosal dissection (ESD)and endoscopic mucosal resection (EMR). However, early stage esophageal squamous cell cancer (ESCC) is highly heterogeneous. M1 and M2 tumors with minor lymph node metastases are suitable for ER. However, the risk of nodal metastases in M3 tumors has been reported to be as high as 18% [6], and all submucosal tumors have a high risk of lymph node metastases [7,8]. Many tumor features have been identified as risk factors for lymph node metastases, including the depth of invasion, tumor size, presence of lymphovascular invasion, etc. [9]. Therefore, superficial ER alone is inadequate for such patients, and additional therapies, including radiotherapy (RT) and concurrent chemoradiotherapy (CCRT), have merited further investigation. It is important to consider the risk of nodal metastasis and the inevitability of recurrence regarding the treatment of early stage ESCC patients. A randomized trial indicated that the 3-year recurrence-free survival (RFS) was 100% in the ESD + RT group and 85.3% in the ESD alone group (*p* = 0.04) [10]. Another pilot study, including 24 T1b-SM2 ESCC patients receiving ER and adjuvant RT, found the local control rate was 100.0%, and the 1-, 3- and 5-year overall survival (OS) rates were 100.0%, 87.4% and 75.0%, respectively [11].

Esophagectomy is still the preferred treatment for T1b and intramucosal M3 patients with lymphovascular invasion. Although esophagectomy has a high cure rate for early stage ESCC, treatment complications and loss of esophagus preservation are concerns. Hence, non-surgical, organ-preserving RT/CCRT has been reported to be a promising treatment alternative to esophagectomy, especially for those who cannot tolerate or are unwilling to receive surgery. A retrospective comparative study of esophagectomy and chemoradiotherapy (CRT) was performed in T1b esophageal cancer patients, and no statistically significant difference in OS was observed [12]. Retrospective or prospective non-randomized controlled studies showed that CRT was comparable or non-inferior to surgery in terms of OS, with the surgery group having relatively more cases of mortality [13,14]. A recent comparative study of 68 patients with T1bN0M0 found that the 5-year RFS rate of the CRT group was significantly lower than the surgery group, but long-term survival could still be achieved by salvage therapy for patients with local recurrence [15].

To the best of our knowledge, few studies have reported detailed outcomes analysis of RT as the definitive or adjuvant modality for different subtypes of T1N0M0 staged patients. We hypothesized that definitive or adjuvant RT/CCRT could be an efficacious treatment of superficial esophageal cancer, especially for patients who are not suitable for ER/surgery or for those with risk factors following ER. Our current study aims to report the outcomes for each group (ER + RT/CCRT and RT/CCRT) with a comparison of outcomes and safety to the existing literature.

## 2. Materials and Methods

### 2.1. Patients

A total of 55 patients with clinical T1N0M0 staged ESCC, treated with ER + RT/CCRT or RT/CCRT in our department from January 2011 to June 2021 were included in this study. Treatment conditions were as follows: 15 patients (27.3%) received ER + RT, 3 patients (5.5%) received ER + CCRT, 11 patients (29.7%) received RT alone and 26 patients (70.3%) received CCRT. The inclusion criteria were as follows: (I) clinical T1N0M0 staged ESCC, (II) an expected minimum life expectancy of more than 6 months, (III) good performance status (PS) (0–2), (IV) baseline laboratory indicators: white blood cell count greater than 2.0 × 10^9^/L; hemoglobin level greater than 80 g/L; platelet count greater than 50 × 10^9^/L; aspartate transaminase/alanine aminotransferase level equal to or less than two times the upper limit of normal; and no severe cardiac abnormalities, (V) patients with high-risk factors including vascular invasion, positive/close margin or submucosal invasion were enrolled in the ER + RT/CCRT group, (VI) patients who were not surgical candidates or refused surgery were enrolled in the RT/CCRT group. The exclusion criteria included patients with other concurrent malignancies, serious medical conditions such as infection and diabetes that were difficult to control and pregnancy or lactation.

All patients were pathologically diagnosed via biopsy with staging confirmed based on the computed tomography (CT) scan or positron emission tomography/computed tomography scan, esophagogastroduodenoscopy with Lugol staining and endoscopic ultrasonography (EUS). Postoperative pathology was classified as T1a or T1b.

### 2.2. Treatment

RT was performed by intensity-modulated RT or volumetric-modulated arc therapy. All patients receiving ER were treated with ESD. The gross tumor volume-tumor bed (GTV-tb) was defined as the involved esophagus between the superior and inferior borders of the ER. When no residual lesion was visible, we requested endoscopists to place 1 or 2 metal clips at the superior and inferior edges of the primary tumor for clinical target volume-tumor bed (CTV-tb) delineation. Gross tumor volume-tumor (GTV-t) was defined as the gross tumor or the region with a positive margin. The clinical target volume (CTV) was defined as the CTV-tb or GTV-t plus a 3 cm margin craniocaudally and 0.7 or 1.0 cm radially, respectively, as well as elective nodal regions, which included supraclavicular fossa and upper mediastinal areas based on the location of the primary tumor. The planning target volume (PTV) was created with a 0.5 cm margin from GTV-t and a 1.0 cm margin from the CTV, which were named as planning gross target volume-tumor (PGTV-t) and PTV, respectively. In the ER + RT/CCRT group, a median dose of 50.4 Gy (range, 50.0–50.4 Gy) was delivered to the PTV. A dose of 59.4 Gy was delivered to the PGTV-t for positive margin patients and 56.0 Gy for close margins. In the RT/CCRT group, a median dose of 50.4 Gy (range, 45.0–54.0 Gy) and 59.4 Gy (range, 55.0–63.6 Gy) was delivered to PTV and PGTV-t, respectively. Definitive RT was defined as RT alone or CRT with an RT dose of more than 50.0 Gy. Adjuvant RT was defined as RT within 4 weeks after ER for patients to prevent a recurrence.

In the RT/CCRT group, 11 patients (29.7%, 11/37) received RT alone due to advanced age, comorbidity or patient refusal. The most common chemotherapy regimens included docetaxel (DTX 75 mg/m^2^/d on days 1 and 21) combined with cisplatin (CDDP, 75 mg/m^2^/d on days 1 and 21)/nedaplatin (NDP, 80mg/m^2^/d on days 1 and 21), CDDP (75 mg/m^2^/d on days 1 and 21)/NDP (80 mg/m^2^/d on days 1 and 21) + capecitabine (1000 mg/m^2^ PO, BID, days 1 to 14, 2 cycles) and CDDP (75 mg/m^2^/d on days 1 and 21)/NDP (80 mg/m^2^/d on days 1 and 21) + S-1 (70 mg/m^2^/day PO, BID on days 1 to 14, 2 cycles). Patients 70 years or older were provided with either capecitabine or S-1 alone after full assessment.

### 2.3. Patient Follow-Up and Endpoint

Patient follow-up was every 3 months for 2 years post-treatment and every 6 months thereafter, with items including physical examinations, complete blood cell count, blood chemistry profiles, CT of the neck, chest, abdomen, etc., performed. We recommended that patients undergo an endoscopy every 6 months to 1 year or when recurrence was clinically suspected. Local recurrence-free survival (LRFS) was measured from the beginning date of RT to the date of first local recurrence or death from any cause. The OS was defined as the period from the beginning of RT to patient death by any cause. Adverse events were classified according to the National Cancer Institute Common Terminology Criteria for Adverse Events (NCI CTCAE, version 4.0).

### 2.4. Statistical Analysis

For continuous variables, we used median and range to indicate their centralized or discrete trend. The Wilcoxon Rank Sum test was used to compare the difference between the two groups for consistency due to its wide applicability. Categorical variables are shown by frequency and percentage. We used the Pearson Chi-square test or continuity correction Chi-square test to determine the differences between the two groups.

Kaplan–Meier survival analysis was used to calculate rates of 1-, 2-, 3- and 5-year LRFS and OS. The Log-rank test was used to evaluate the difference in the time-to-event outcomes between the different tumor stage groups. The Statistical Package for Social Sciences (SPSS version 24.0, IBM, Armonk, NY, USA) software was used for statistical analyses. A two-tailed *p*-value < 0.05 was considered to be statistically significant.

## 3. Results

### 3.1. Patient Characteristics

Collected patient information is listed in Table 1. The median age was 69 years old, and the median pre-treatment tumor length measured by endoscopy was 3.0 cm in all patients. Details of the ER + RT/CCRT group (*n =* 18) and the RT/CCRT group (*n =* 37) are also displayed. The median interval time between ER and RT/CCRT was 44 days. Compared with the ER + RT/CCRT group, the patients in the RT/CCRT group were significantly older (*p* = 0.002), had a significantly longer median tumor length (*p* = 0.017), had significantly worse Karnofsky Performance Status (KPS) (51.5% vs. 83.3% in KPS = 90, *p* = 0.022) and less patients had T1b disease (27.0% vs. 88.9%, *p* < 0.001). 

Reasons for receiving the different treatment modalities are outlined in Table 2. In the ER + RT/CCRT group, all patients harbored risk factors including submucosal invasion (88.9%, 16/18, three of them also had a positive margin, and one also had a close margin) or vascular invasion (11.1%, 2/18). In the RT/CCRT group, 34 patients (91.9%, 34/37) were not viable surgical candidates, and 3 patients (8.1%, 3/37) refused surgery.

### 3.2. Survival Analysis

Median follow-up time was 51.9 months (range 4.5–121.3 months), and six patients (10.9%, 6/55) had died at the time of the last follow-up. The estimated LRFS at 1-, 2-, 3- and 5 years was 92.7%, 92.7%, 90.7% and 90.7%, respectively (Figure 1). The estimated OS rates at 1-, 2-, 3- and 5 years were 96.4%, 94.4%, 92.4% and 92.4%, respectively (Figure 1). The estimated 5-year LRFS and OS in the ER + RT/CCRT group was 88.9% (95%CI: 74.4–100.0%) and 94.4% (95%CI: 83.8–100.0%) (Figure 2a). The estimated 5-year LRFS and OS in the RT/CCRT group was 91.8% (95%CI: 83.0–100.0%) and 91.7% (95%CI: 82.7–100.0%), respectively (Figure 2b).

The Log-rank test of LRFS and OS in the two group showed no significant survival difference among different tumor stage in terms of LRFS and OS in the ER + RT/CCRT group (*p* = 0.611, *p* = 0.724, respectively) and the RT/CCRT group (*p* = 0.645, *p* = 0.641, respectively).

### 3.3. Failure Patterns

The failure patterns, including any local or regional recurrence, metastasis and death, are shown in Table 3. In-field local failure occurred in three patients (5.5%, 3/55), with one patient in the ER + RT/CCRT group and two patients in the RT/CCRT group. The patient with in-field local failure in the ER + RT/CCRT group underwent salvage ESD and achieved complete pathological remission. The other two patients in the RT/CCRT group did not receive any salvage therapy and died 1 and 2 months after recurrence. In the RT/CCRT group, pulmonary metastasis was observed in one patient (1.8%, 1/55) 60 months after treatment. No patients developed regional recurrence. The causes of death (six patients) included local recurrence (two patients), lung metastasis (one patient), infectious pneumonia (two patients), and one patient died of myocardial infarction without heart disease prior to RT.

### 3.4. Toxicities

The most common toxicities were Grades 1–2 in all patients, including esophagitis (74.5%, 41/55), myelosuppression (49.1%, 27/55) and esophageal stricture after RT (27.3%, 15/55) (Table 4). Within the ER + RT/CCRT group, seven patients (38.9%, 7/18) developed Grade 1–2 esophageal stricture after RT, and two patients (11.1%, 2/18) developed Grade 3. Eight patients (21.6%, 8/37) and four patients (10.8%, 4/37) developed Grade 1–2 and Grade 3 esophageal stricture after RT in the RT/CCRT group, respectively. One patient with Grade 3 underwent subsequent endoscopic balloon dilation in the RT/CCRT group. In the diet follow-up survey, among 55 patients, 48 patients (87.3%) were capable of maintaining a regular diet, 6 patients (10.9%) were on a soft or semifluid diet and 1 patient (1.8%) had a full liquid diet due to dysphagia. Grade 1–2 pericardial effusion occurred in two (11.1%, 2/18) and four (10.8%, 4/37) patients in the ER + RT/CCRT and RT/CCRT groups, respectively. No patients experienced Grade 4 or higher toxicity, and there was no treatment-related death.

## 4. Discussion

The current study showed that definitive or adjuvant RT for superficial ESCC patients resulted in satisfactory clinical outcomes and toxicities. The LRFS and OS at 1-, 2-, 3- and 5 years are similar to previous reports [15,16,17,18] and were comparable to early esophageal patients treated with surgery, with the 5-year OS ranging from 77.7% to 85.5% [14,15,19]. Therefore, we find RT as an effective treatment alternative for superficial ESCC.

According to previous reports [20], after ER, 1-, 3- and 5-year survival rates for pT1a patients were 98.5%, 91.6% and 86.3%, and 1-, 3- and 5-year survival rates for T1b patients were 93.3%, 77.6% and 68.9%. Superficial esophageal M3 and T1b tumors have a relatively higher risk of lymph node metastases [6,7,8]. Therefore, it has become necessary to consider the limitations of ER in the context of nodal metastasis risk and inevitability of recurrence for early ESCC patients. For such patients, the addition of RT deserves further investigation. In this study, the 5-year OS rate was 94.4% in the ER + RT/CCRT group, which was consistent with other studies. Shimizu et al. estimated a 100% 5-year OS rate in their ER + CCRT group [17]. Yang et al. [16] reported 1-, 3- and 5-year OS rates of 100.0%, 86.9% and 68.5% for cT1N0M0 staged patients who underwent ER + RT. A retrospective study compared two types of additional therapy (surgery or CRT) after ER in patients with pT1b (SM) disease. The 3- and 5-year OS of the CRT group was 90.4% and 80.3%, respectively [21].

It is possible that ER + RT/CCRT could reduce the risk of local, regional or distant metastasis, thus improving survival outcomes. A randomized control trial showed that the 3-year RFS was 100% in the ESD + RT group and 85.3% in the ESD alone group (*p* = 0.04) [10]. Another pilot study including 24 T1b-SM2 ESCC patients receiving ER and adjuvant RT indicated the local control rate was 100.0%, with 1-, 3- and 5-year OS rates of 100.0%, 87.4% and 75.0% [11]. Yang et al. [16] found that additional RT following ER achieved local and regional control rates of 100.0% and 93.5%, respectively. Osamu et al. [22] indicated that RT following ESD improved the local control rate in pT1a mm and pT1b-SM1 ESCC patients. In our study, the 5-year LRFS was 88.9%, and no regional or distant metastasis was observed in the ER + RT/CCRT group. Although clinical outcomes are encouraging, recurrence is still a concern for the ER + RT modality in relation to radical surgery. In Tanaka et al. [21], there was no recurrence in the ER + surgery group for patients with pT1b esophageal cancer receiving surgical resection after ER, but six recurrences occurred in the ER + RT group. Of the six patients with recurrence, four patients were administered CCRT with only 41.4 Gy. Their univariate analysis suggested that positive lymphatic invasion was the significant risk factor for recurrence in the CCRT group. Similarly, Ikawa et al. [23] showed that 7 (7.3%) of 96 ER + RT patients had regional lymph node recurrence, which was higher than that reported by Uchinami et al. (1 of 71 patients, 1.4%) [24]. This difference may, in part, be explained by the higher doses used in Uchinami’s study (39.6–50.4 Gy) than those in Ikawa’s study (40.0–41.4 Gy). In our study, patients treated with ER + RT/CCRT had risk factors, including submucosal invasion or vascular invasion. Only one (5.6%, 1/18) recurrence and no node metastasis occurred in the 18 patients. A median dose of 50.4 Gy (range, 50.0–50.4 Gy) was delivered to the PTV, and 59.4 Gy was delivered to PGTV-t for positive margin and 56.0 Gy to close margins, respectively. Therefore, we suggest a total dose greater than or equal to 50.0 Gy may be more suitable for local and regional control for T1 patients with risk factors receiving ER + RT.

The standard treatment modality is generally considered to be radical esophagectomy for superficial esophageal [25]; however, postoperative complications after esophagectomy may reduce the patient’s quality of life and long-term survival rate [26]. In this study, the 5-year OS and LRFS of the definitive RT/CCRT group were 91.7% and 91.8%, respectively. Haneda et al. compared prognosis between surgery and definitive CRT in 68 patients with T1bN0M0 ESCC and no significant difference was found in OS between the two groups. However, recurrence remained a concern, with the 5-year RFS rate in the surgery group being significantly higher than that in the definitive CRT group (91.1% vs. 62.7%, *p* = 0.039) [15]. Another study [27] showed that locoregional failure occurred in 26% of the CCRT group compared to none in the ER + CCRT group (*p* < 0.01). Five-year relapse-free survival in the ER + CCRT group was significantly better than that in the CCRT group (85.1% vs. 59.2%, *p* < 0.05). In our study, two patients in the RT/CCRT group had in-field recurrence. However, it was shown that salvage therapy for local recurrence could improve survival [15]. Haneda et al. [15] also showed no disease progression in patients receiving endoscopic submucosal dissection or photodynamic salvage therapy following local recurrence after definitive CCRT. In this study, the two recurrent patients ultimately died without salvage treatment due to refusing treatment after recurrence.

In our study, the most common toxicities were esophagitis, myelosuppression and esophageal stricture after RT in both groups. The toxicity rate in the two groups was similar to previous reports [28,29]. No grade 4 reactions were observed. Several previous studies showed that patients with ≥3/4 of the esophageal circumferential invasion and a long longitudinal diameter could have a high rate of postoperative esophageal stricture [15,28,30]. Kawaguchi et al. showed that 75% (3/4) of patients had esophageal stenosis before CRT after ER, and esophageal stenosis did not worsen after CRT. Furthermore, the incidence of esophageal stenosis in the CRT group was only 3% [31]. In addition, clinical trials frequently exclude elderly patients. While in our study, 15 patients were 70 years old or older, with 2 receiving ER + RT/CCRT and 13 receiving RT/CCRT. The results show that RT/CCRT is safe for elderly early esophageal patients, which is consistent with other reports [31]. Often, young and fit patients are ideal surgical candidates, while the elderly and unfit receive conservative treatments, partially explaining “early deaths”. Furthermore, mortality at 90 days after esophagectomy is between 6–10%, even after minimally invasive resection [32], indicating RT is an alternative for early stage ESCC patients with acceptable toxicities. A summary of previous studies and their findings discussed in the current report can be found in Appendix A.

The present study had several limitations. First, due to the retrospective nature of the current study, the results could be impacted by potential confounder biases. Second, the sample size was relatively small, and the single institutional and solely squamous histology nature may limit its ability to generalize these results to a larger population, especially those with adenocarcinoma histology. In addition, definitive radiotherapy and ER followed by adjuvant radiotherapy are more commonly applied in Asia for early stage esophageal cancer (EC), and surgery is still recommended by the National Comprehensive Cancer Network (NCCN) guidelines [33]. Nevertheless, the Japan Clinical Oncology Group (JCOG) 0508 trial confirmed that adjuvant chemoradiotherapy after ER is as effective as surgery [34]. Therefore, the treatment modality of definitive or adjuvant radiotherapy is worth further evaluation in squamous EC patients in the western world. Moreover, the differences in practice among different institutions and even different countries could result in different clinical outcomes, likely limiting the generalizability of findings in the current study. Third, some clinical data was not recorded, such as the description of patients with >3/4 of the esophageal circumferential invasion in relation to the post-operational esophageal stricture. Furthermore, some patients with T1 disease could not be clearly defined as T1a or T1b. A multicenter and prospective study is needed to comprehensively determine the role of RT in early stage esophageal cancer.

## 5. Conclusions

In conclusion, definitive or adjuvant RT for superficial ESCC patients showed satisfactory clinical outcomes and acceptable toxicities. Radiotherapy is an effective alternative treatment method for superficial ESCC compared to surgery.

## Figures and Tables

**Figure 1 cancers-14-03423-f001:**
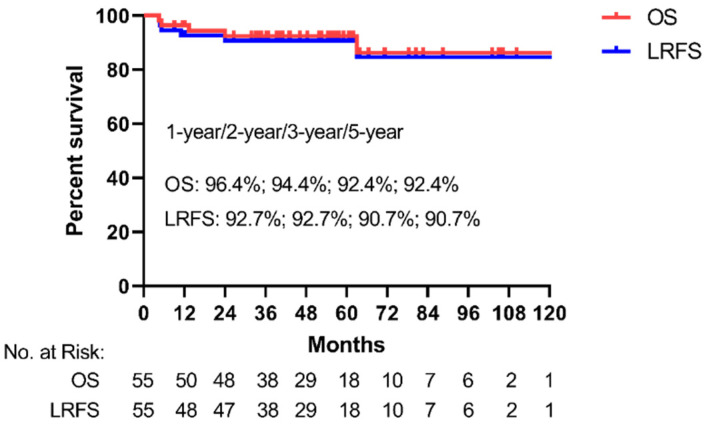
Kaplan–Meier analysis of local recurrence-free survival (LRFS) and overall survival (OS) of all patients. The percent survival of 1-, 2-, 3- and 5-year of LRFS and OS rate and the number at risk are also presented.

**Figure 2 cancers-14-03423-f002:**
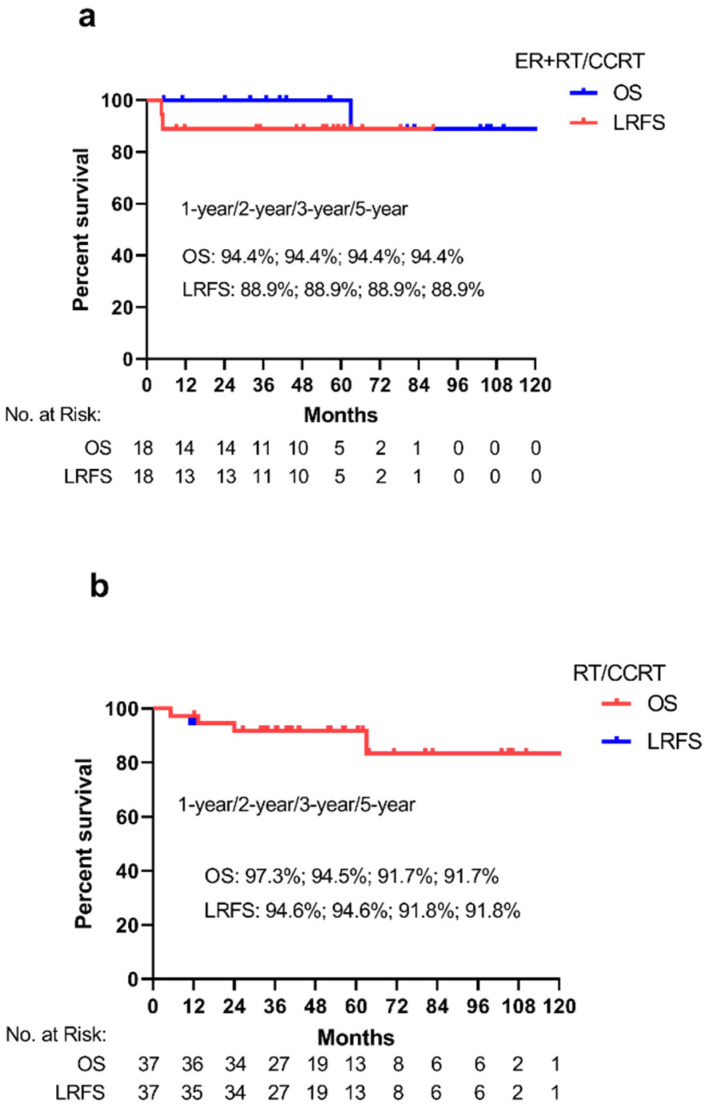
Kaplan–Meier analysis of local recurrence-free survival (LRFS) and overall survival (OS) in two groups: ER + RT/CCRT group (**a**); RT/CCRT group (**b**). The percent survival of 1-, 2-, 3- and 5-year of LRFS and OS rate and the number at risk are also presented.

**Table 1 cancers-14-03423-t001:** The characteristics of patients and tumors in the two groups [M (range)/n (%)] ^†^.

Variables	Total	ER + RT/CCRT (*n =* 18)	RT/CCRT (*n =* 37)	*p*
Age (years)	69.0 (51.0, 83.0)	62.0 (51.0, 74.0)	69.0 (52.0, 83.0)	0.002
Tumor length in EUS (cm)	3.0 (1.0, 10.0)	2.0 (1.0, 4.0)	3.5 (1.0, 10.0)	0.125
Tumor length in gastroscope (cm)	3.0 (1.0, 18.0)	2.0 (1.0, 7.0)	4.0 (1.0, 18.0)	0.017
Tumor length in GTV-t (cm)	7.5 (4.0, 19.0)	-	7.5 (4.0, 19.0)	-
Tumor length in GTV-tb (cm)	8.3 (4.5, 10.5)	8.3 (4.5, 10.5)	-	-
PGTV-t dose (Gy)	59.4 (55.0, 63.6)	59.4 (56.0, 59.4)	59.4 (55.0, 63.6)	0.730
PTV dose (Gy)	50.4 (45.0, 54.0)	50.4 (50.0, 50.4)	50.4 (45.0, 54.0)	0.742
The interval between ER and RT/CCRT(day)	44 (20, 66)	44 (20, 66)	-	-
Sex				
Male	39 (70.9)	14 (77.8)	25 (67.6)	0.434
Female	16 (29.1)	4 (22.2)	12 (32.4)	
Basic diseases				
None	37 (58.7)	14 (73.7)	23 (52.3)	0.467
Hypertension	8 (12.7)	1 (5.3)	7 (15.9)	
COPD	3 (4.8)	0(0)	3 (6.8)	
Diabetes	5 (7.9)	2 (10.5)	3 (6.8)	
Cardiovascular and cerebrovascular diseases	7 (11.1)	1 (5.3)	6 (13.6)	
Others	3 (4.8)	1 (5.3)	2 (4.5)	
Alcohol abuse				0.196
No	37 (67.3)	10 (55.6)	27 (73.0)	
Yes	18 (32.7)	8 (44.4)	10 (27.0)	
Smoke				0.214
No	28 (50.9)	7 (38.9)	21 (56.8)	
Yes	27 (49.1)	11 (61.1)	16 (43.2)	
KPS				0.022
80	21 (38.2)	3 (16.7)	18 (48.6)	
90	34 (61.8)	15 (83.3)	19 (51.5)	
Single or multiple primary tumors				0.943
Single	44 (80.0)	15 (83.3)	29 (78.4)	
Multiple	11 (20.0)	3 (16.7)	8 (21.6)	
Tumor stage				<0.001
Tis	3 (5.5)	0 (0)	3 (8.1)	
T1a	9 (16.4)	2 (11.1)	7 (18.9)	
T1b	26 (47.3)	16 (88.9)	10 (27.0)	
Unknown T1 substage	17 (30.9)	0 (0)	17 (45.9)	
Main tumor location				0.653
Upper thoracic	4 (7.3)	2 (11.1)	2 (5.4)	
Middle thoracic	35 (63.6)	10 (55.5)	25 (67.6)	
Lower thoracic	16 (29.1)	6 (33.3)	10 (27.0)	

^†^ The data was shown as Median (range) or number (percent).

**Table 2 cancers-14-03423-t002:** The reasons for receiving different treatment modalities.

T Stage	ER + RT/CCRT (*n =* 18)		RT/CCRT (*n =* 37)	
	Reasons	*n*	Reasons	*n*
Tis (*n =* 3)		0	Not suitable for surgery	2
			Refused surgery	1
T1a (*n =* 9)	Vascular invasion	2	Not suitable for surgery	7
T1b (*n =* 26)	Submucosal invasion with a positive margin	3	Not suitable for surgery	8
	Submucosal invasion with a close margin	1	Refused surgery	2
	Submucosal invasion	12		
Unknown T1 substage (*n =* 17)		0	Not suitable for surgery	17

**Table 3 cancers-14-03423-t003:** The failure patterns [*n* (%)] ^†^.

Failure Patterns	Total (*n* = 55)	ER + RT/CCRT (*n* = 18)	RT/CCRT (*n* = 37)
In field local failure	3 (5.5)	1 (5.6)	2 (5.4)
Out-field local failure	0 (0)	0 (0)	0 (0)
Regional recurrence	0 (0)	0 (0)	0 (0)
Metastasis	1 (1.8)	0 (0)	1 (2.7)
Death	6 (10.9)	1 (5.6)	5 (13.5)

^†^ The data was shown as number (percent).

**Table 4 cancers-14-03423-t004:** Toxicities [*n* (%)] ^†,‡^.

Toxic Effects	Total (*n* = 55)		ER + RT/CCRT (*n* = 18)		RT/CCRT (*n* = 37)	
	Grade 1–2	Grade 3	Grade 1–2	Grade 3	Grade 1–2	Grade 3
Acute						
Esophagitis	41 (74.5)	4 (7.3)	10 (55.6)	2 (11.1)	31 (83.8)	2 (5.4)
Myelosuppression	27 (49.1)	2 (3.6)	9 (50.0)	1 (5.6)	18 (48.6)	1 (2.7)
Radiation pneumonitis	10 (18.2)	0 (0)	4 (22.2)	0 (0)	6 (16.2)	0 (0)
Esophageal fistula	0 (0)	0 (0)	0 (0)	0 (0)	0 (0)	0 (0)
Late						
Esophageal stricture	15 (27.3)	6 (10.9)	7 (38.9)	2 (11.1)	8 (21.6)	4 (10.8)
Pericardial effusion	6 (10.9)	0 (0)	2 (11.1)	0 (0)	4 (10.8)	0 (0)

^†^ The data was shown as number (percent). ^‡^ Toxicity was graded according to the National Cancer Institute Common Terminology Criteria for Adverse Events (NCI CTCAE v 4.0).

## Data Availability

The data presented in this study are available on request from the corresponding author. The data are not publicly available due to patient privacy.

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
