# Peer review of "Long-Term Clinical Outcomes and Safety Analysis of Superficial Esophageal Cancer Patients Treated with Definitive or Adjuvant Radiotherapy"

_cancers, 2022, doi:10.3390/cancers14143423_

Round 1
Reviewer 1 Report
The report presents data on an interesting topic. However there are some english spellings and sentences that have to be improved. I would advice to ask a native english speaker to improve the spelling and wording
Some examples: Line 62, 117, 181 (risky factors --> risk factors), line 220, 292, 313, 329, 335
"Patient characteristics" is not a result and should be transferred to the patients and methods section.
Figure 4 can be omitted. Describe in tect
Reviewer 2 Report
Review of the manuscript :
« long term clinical outcomes and safety analysis of superficial esophageal cancer patients treated with definitive or adjuvant radiotherapy »
Lyu et al. Cancers 1773588
This paper is focused on outcomes of patients treated medically for clinical T1N0M0 esophageal squamous cell carcinoma, data extracted from their series of 55 patients treated between 2011 and 2021.
Minor comments
Line 81 : say a « retrospective » comparative study …
Line 87 : please reword this sentence currently not clear
Line 104 : or submucosal « invasion » ?
Line 146 : please describe a little bit more endoscopic follow-up, was it really /3 months untill year 2 then / 6 months ?
Line 171 : I do not understand : « the median tumor lenght examined by gastroscope was 3.0 cm », carefully examined, previously resected and carefully examined ? please reword
Table 1 : not easy to read, reorganize please
Table 1 : replace « drink » by alcohol abuse ( ?)
Table 2 : Refuse to surgery : refuse surgery ? or contra-indicated for surgery ?
When was observed the lung metastase : early or late ?
Line 259 : it’s – it is
Major comments
Table 2 : not very useful, I think ; but this table shows that, certainly, many patients who had ER did not have CCRT, some data about this group may be of interest (but it is up to you) ;
Fig 2b : I can’t see the blue curve ; legend : … in the 2 groups.
Globally I am not sure that most of your Kaplan-Meier curves are of interest taking into account the low number of patients ;
Line 222 : and eventually died … please replace eventually by real data, it is unacceptable as now.
In the Discussion part please insist on the fact that previous ER did not increase the risk of post radiotherapy stenosis.
Conclusion
Paper of interest, deserves publication.
A little bit too long ; to shorten particularly the discussion ; most figures may be deleted.
Regarding the OS data it is of importance to say that, in many centers, young and fit patients may be operated on and only old and unfit had conservative treatment explaining part of the « early deaths ».
It will be also of interest to say that 90 days mortality after esophagectomy is between 4 and 6-10%, even after minimally invasive resection and to put that in perspective. (Mariette C, N Engl J Med 2019)
Reviewer 3 Report
Thank you for the opportunity to review this manuscript. The study is interesting. Below please see my comments.
1. Please provide clear definitions for definitive vs. adjuvant radiotherapy. Readers may not be familiar with these terms. Having clear definitions of these terms are important for readers to understand the goals, study design, and findings of this study. Please also see my comment #2.
2. The aims of the study in the introduction could be made more clear. It will be helpful to clearly state that the goals are not to compare between the two treatment groups but to report the outcomes for each group separately, ET+RT/CCRT (adjuvant?) vs. RT/CCRT (definitive?). To determine if these treatments have “favorable” outcomes, the comparison of outcomes and safety were made to previous literature. This was not immediately clear to me when reading the article, partly because the definitions for definitive vs. adjuvant radiotherapy were not provided upfront. The description of the methods gives the impression that the goals were to compare the two treatment groups. (e.g. see my comment #8).
3. Abstract: Please spell out LRFS and OS.
4. The statement “The most common toxicity were Grade 1-2 in all patients, including esophagitis (74.5%, 41/55), myelosuppression (49.1%, 27/55) and esophageal stricture after radiotherapy (27.3%, 15/55), respectively. Two patients (11.1%, 2/18) and 4 patients (10.8%, 44 4/37) had Grade 3 esophageal stricture after RT in the ER+RT/CCRT group and RT/CCRT group, 45 respectively.” This statement is not consistent with the findings in Table 4. There are Grade 3 esophageal strictures reported. Also the number of patients are not consistent. This same inconsistency was also found in the results section above Table 4. Please check which ones are correct.
4. Please clearly explain the difference between ER and ESD for general readers.
5. It is important to see the differences between the groups. Suggest to combine Table 1 and Table S1 and include as Table 1 in the main text. That is, having columns by group and p-values and a separate column of the total.
6. Due to the small sample sizes, especially the ER+RT/CCRT group (only 18), subgroup analyses within each treatment group may not be conclusive. For instance, in Figure 3, fairly large differences are seen by stage, however, the p-value was not statistically significant. One could not conclude from this that OS does not differ by stage because of the potentially insufficient power to test these differences.
7. Line 150, “local recurrence-free survival was … from the beginning date of RT to date of first local recurrence..” Not all patients had RT. What is the starting date of followup for those without RT?
8. Statistical analysis section, lines 163, “log-rank test was used to test the differences between the groups”. Which groups? This gives the impression that the two treatment groups are compared. But they were not (Figure 1 reported separate KM curves.). Please be clear which groups are compared and tested using log-rank test. Please also see my comment #1 and #2.
9. The discussion of the findings from previous studies are hard to follow. Could the authors create a table summarizing the findings from previous studies and their findings?
10. Please provide a clear definition of what is considered a “failure”. A failure of treatment?
11. The authors concluded that the findings showed favorable clinical outcomes. Favorable compared to what? Please revise the wording to clear state what the authors mean instead of the word “favorable” which suggest a comparison, yet no direct comparison was made in this study (e.g. compare to surgery group).
12. Also, although the numbers are compared to previous studies, there were no test of statistically significantly differences in OS or LRFS from previous findings. The small sample sizes of this study also cast some doubt whether the findings are unique to this sample population and could be generalized. The discussion should be expanded in the limitation section.
Round 2
Reviewer 3 Report
I would like to thank the authors for taking the time to address my concerns. I have only one minor comment.
1. My last comment about the generalizability was not to say that because it is small sample size that it is not generalizable. Even if the sample size is large, it could still be limited in generalizablity if there are significant institutional or country differences . The authors should consider if there are any reasons why these findings are not likely to generalize to other patients: e.g. Are patients included unique? Could the clinical practice in China differs from to other countries? Would the treatment to be different by countries or settings?
